# New Psychoactive Substances: Awareness and Attitudes of Future Health Care Professionals in Serbia

**DOI:** 10.3390/ijerph192214877

**Published:** 2022-11-11

**Authors:** Vesna Mijatović Jovin, Nina Skoko, Ana Tomas, Dejan Živanović, Darija Sazdanić, Nemanja Gvozdenović, Ana-Marija Vejnović

**Affiliations:** 1Department of Pharmacology, Toxicology and Clinical Pharmacology, Faculty of Medicine, University of Novi Sad, 21000 Novi Sad, Serbia; 2Department of Biomedical Sciences, College of Vocational Studies for the Education of Preschool Teachers and Sports Trainers, 24000 Subotica, Serbia; 3Department of Psychology, College of Social Work, 11000 Belgrade, Serbia; 4Department of Pharmacy, Faculty of Medicine, University of Novi Sad, 21000 Novi Sad, Serbia; 5Department of Emergency Medicine, Faculty of Medicine, University of Novi Sad, 21000 Novi Sad, Serbia; 6Clinical for Orthopedic Surgery and Traumatology, Clinical Center of Vojvodina, 21000 Novi Sad, Serbia; 7Department of Psychiatry and Psychological Medicine, Faculty of Medicine, University of Novi Sad, 21000 Novi Sad, Serbia; 8Clinic for Psychiatry, Clinical Center of Vojvodina, 21000 Novi Sad, Serbia

**Keywords:** new psychoactive substances, awareness, attitudes, students, health care professionals

## Abstract

This study was conducted in order to evaluate prospective health care professionals’ (HCPs) awareness and attitudes regarding new psychoactive substances (NPSs) in the context of their future role in the prevention and treatment of NPS overdose and addiction. Correlates of NPS perception and use were also examined. This cross-sectional survey was performed on 490 students of the Faculty of Medicine, Novi Sad, Serbia, during 2017. NPS awareness was better in pharmacy students (IRR: 1.926, CI: 1.173–3.163, *p* = 0.010) than in medicine students—pharmacy students recognized 92.6% more NPS names than their peers studying medicine. Female students knew 36.5% less NPS names than their male colleagues (IRR: 0.635, CI: 0.399–1.013, *p* = 0.049). Number of NPS names students knew was rising by 15.9% with each age group—the higher the age, the larger the number of NPSs they were aware of (IRR: 1.159, CI: 1.025–1.310, *p* = 0.018). Students who had used marijuana knew 52.6% more NPS names than those who had never had experience with cannabis (IRR: 1.526, CI: 0.953–2.445, *p* = 0.049). Although a high number of future HCPs claimed to know what NPSs are, numerous misconceptions were noticed. Further educational efforts are necessary to improve their awareness and attitudes regarding NPSs.

## 1. Introduction

The United Nations Office for Drugs and Crime has defined NPSs (new psychoactive substances) as ‘substances of abuse, either in a pure form or a preparation, that are not controlled by the 1961 Single Convention on Narcotic Drugs or the 1971 Convention on Psychotropic Substances, but which may pose a public health threat’ [1]. According to the distinguished experts in the field, NPSs are also defined as narcotic drugs or psychotropic substances made available or used from the early to mid-2000s for their psychoactive properties [2].

NPSs can be classified according to their effects (hallucinogens, stimulants, or depressants), their origin (natural, synthetic, or semisynthetic), legal situation (lawful, illicit, or unregulated), or chemical structure (polycyclic hydrocarbons, amines, alcohol/ethers, and other NPSs) [2,3,4]. By their number, nature, and composition, NPSs represent unprecedented challenges not only for drug consumers but also for clinicians, researchers, toxicologists, drug control policies, and other health care and regulatory bodies worldwide [3].

In Serbia, the appearance of NPSs can be traced back to 2010-11, while their expansion dates from the appearance of synthetic cannabinoids and their legal availability through “Head Shops” from 2013-14 [5]. Estimation on the prevalence of NPS use is very difficult to obtain in Serbia. Data are scarcely available in analyses of calls to national poison control centers, emergency department admissions, and drug use surveys. According to the annual reports of the Serbian Poison Control Center, a large increase in the number of acute poisonings with NPSs was registered for the first time in 2015—8.4% of poisonings caused by psychoactive substances were diagnosed as acute poisoning with synthetic cannabinoids [6]. In addition, the Early Warning System in Serbia has been active since 2016 and an increase in the prevalence of NPS use has been reported within European Monitoring Centre for Drugs and Drug Addiction publication next year [7]. Moreover, just two reliable publications in international peer-reviewed journals are available regarding NPSs in Serbia. The first one pointed out clinical and analytical experience of the Serbian National Poison Control Centre with synthetic cannabinoids [8], while the second one examined the use of NPSs in light of harm-reduction responses in six Euroasian countries, including Serbia [9].

Health care provider awareness is recognized as a headstone of public health initiatives in taking up the upcoming threat presented by NPS [10]. Recently published studies have underlined the inadequacy in essential knowledge and skills required by health care professionals (HCPs) to successfully treat patients overdosed by NPSs [11,12]. Medical education including continued education, by all means, has a significant impact on how medical personnel approach substance abuse problems. Numerous investigations have shown insufficient knowledge of prospective doctors regarding not only classic illegal psychoactive substances but also synthetic cannabinoids and medical cannabis [13,14,15,16,17]. As management of NPS toxicity is not dependent on knowing the particular substance but should be led by consumers’ clinical signs and symptoms, physicians are less confident in managing acute NPS toxicity than toxicity provoked by classical recreational drugs [18]. The deficit in general knowledge on NPSs may be responsible for inappropriate medical diagnoses, advice, and/or interventions [19].

We conducted an awareness survey among university students in order to determine if prospective HCPs have sufficient knowledge and understanding about NPSs in the context of their future role in the prevention and treatment of NPS overdose and addiction. This paper is the first, to our knowledge, to provide prevalence estimates, to examine correlates of NPS perception, and to use among future HCPs not only in Serbia but also in the Balkan region.

## 2. Methods

The survey was conducted on a sample of students of integrated academic studies of medicine, pharmacy, and dentistry at the Faculty of Medicine, University of Novi Sad, Serbia, between 20 November and 20 December 2017. The study was approved by the Ethical Committee of the Faculty of Medicine, University of Novi Sad, Serbia.

In Serbia, studies are organized as integrated master studies, with pharmacy and dentistry being a 5-year program and with medicine being a 6-year program.

The questionnaire was created by using a survey by Martinotti et al. (2015) [20], with the modifications necessary to ensure correct answers to questions and claims and to be more suitable for target population—future health care professionals (HCPs).

The students were informed that survey completion was voluntary and confidential and that no personal information would be collected/processed and published. After obtaining consent for participation, the survey was started. Students completed the questionnaire during the last 15 min of their scheduled classes. The content, comprehension, readability, and design of the questionnaire were pre-tested on 20 students.

The questionnaire was divided into three sections. The first one investigated respondents’ socio-demographic and academic characteristics including age, gender, year of study, subject of study (medicine, pharmacy, or dentistry), study duration, parents’ educational level, residence, living status, incomes during studies, and social network account. The second part referred to students’ habits regarding (il)legal psychoactive substances (ab)use (tobacco, alcohol and energy drinks, marijuana, and other “classic” illegal psychoactive substances). The third section examined students’ awareness and potential use of NPS, their reasons for such behavior, effects of consumption, source of information about NPSs, and recommendation for protective and preventive measures. The NPSs included in the questionnaire were synthetic cannabinoids (spices and spice drugs), mephedrone (Meow-Meow, bubbles, and M-Cat), phenethylamines, methoxetamine, methamphetamine, desomorphine, and methylenedioxypyrovalerone (Ivory Wave, Vanilla Sky, Pure Ivory, Purple Wave, Charge+, Ocean Burst, and Sextacy). Students were also encouraged within one open-label question in this part of the questionnaire to write in some additional NPS names.

Statistical analysis was performed with IBM SPSS software (SPSS 22.0 for Windows, SPSS Inc., Chicago, IL, USA). Out of the descriptive statistical methods, measures of central tendency (arithmetic mean), measures of variability (standard deviation), and frequency were used. A Chi-square test was used to test the differences between nominal data (frequencies), while a *t*-test for independent samples and ANOVA with Tuckey post hoc test were used for numerical variables. Mann–Whitney U and Kruskal–Wallis tests were used for numeric data with non-parametric distribution. In order to analyze predictors for NPS awareness among future HCPs, a negative binomial model was applied, while binary logistic regression was administered to analyze the predictors for NPS use. The first predicted outcome was a count of NPS names students know, while the second one was potential use of NPSs among respondents. Predictors for both mentioned outcomes were 1. socio-demographic and academic factors (age, gender, subject of study, size of the place of residence, source of income, with whom students lived, the highest parents‘ educational level, repeating a study year (redoublement), and having a social networking profile) and 2. students’ habits regarding (il)legal psychoactive substances (ab)use (frequency of smoking, energy drink usage, alcohol consumption, getting drunk, and marijuana use). Results were reported as incidence rate ratio (IRR) with 95% confidence intervals (CI). All *p*-values less than 0.05 were considered significant.

## 3. Results

The total number of respondents was 490. Their average age was 21.57 ± 2.04 years, and the majority of them were female (76.73%). Considering the subject and year of study, participants were almost equally distributed (Table 1). One third of the respondents (32.04%) lived with parents, while the rest lived in the rented and own apartments or university dormitories (30.20%, 25.51%, and 11.63%, respectively). The majority of students (90.61%) were financed by parents. Geographical distribution indicated that 80.61% of the sample lived in urban areas. Having a social network profile was reported by almost all the study participants (97.14%).

Half of the subjects (54.29%) declared that they are tobacco smokers. The percentage of energy drinks consumption in the sample was 87.96%. Alcohol was mostly used on special occasions (83.88%), and 76.33% of alcohol consumers had drunk alcoholic beverages until losing control at least once. Marijuana use was admitted by 24.49% of the future HCPs, while the usage of other illegal psychoactive substances was not commonly reported (94.08%). Even though 54.69% of the respondents had a friend who was marijuana consumer, only 25.71% became acquainted with someone who abused hard drugs (Table 2).

Participants’ gender, subject of study, and year of study were found to be significantly associated with students’ habits regarding (il)legal psychoactive substances (ab)use. A significantly higher proportion of dentistry (64.7%) and medicine (55.8%) students than pharmacy students (43.4%, *p* < 0.001) were tobacco consumers. Similarly, the usage of marijuana was reported more frequently among dentistry and medical students (27.97%) in comparison with pharmacy students (17.2%, *p* < 0.036). In addition, male students were more likely to be marijuana users (36.6% males vs. 21.0% females, *p* = 0.01). Taking into account the study year, about 15% of the first year students consumed marijuana at least once, but over 25% in their fourth study year or later (*p* = 0.02) consumed marijuana. Male students drank more often and in larger quantities compared to female students. Namely, 88.4% male students got drunk at least once vs. 71.7% female students (*p* < 0.001), while 20% males vs. 10.35% females had drunk alcoholic beverages until losing control regularly (*p* = 0.02). About 56% of the first-year students got drunk at least once, and 80% students in their fifth and sixth years (*p* = 0.02) got drunk at least once. Students who reported the usage of harder drugs more often consumed alcohol and marijuana (*p* < 0.001).

A quarter of participants (115—23.47%) claimed that they knew what NPSs were. This group also recognized NPS names more easily than the students who asserted that did not know what NPSs were (*p* < 0.001). The most commonly marked NPS names were Vanilla Sky, mephedrone, and methoxetamine, recognized by 34.8%, 24.3%, and 20.9% participants, respectively (Table 3). As students were encouraged to add NPS names to the list, the added items were Angel Dust, Black Mamba, crocodile, White Widow, club drugs, and new synthetics. The main sources of information regarding NPSs were the internet (40.51%), theoretical lectures (23.79%), traditional media (21.86%), friends (12.22%), and practical (clinical) lectures (1.62%).

NPS awareness (estimated by number of NPS names students recognize) was better in pharmacy students (IRR: 1.926, CI: 1.173–3.163, *p* = 0.010) than in medicine students—in other words, pharmacy students knew 92.6% more NPS names than their peers studying medicine. In addition, female students knew 36.5% less NPS names than their male colleagues (IRR: 0.635, CI: 0.399–1.013, *p* = 0.049). Number of NPS names students know was rising by 15.9% with each study year—the higher the age, the larger the number of NPS they were aware of (IRR: 1.159, CI: 1.025–1.310, *p* = 0.018) (Table 4).

As it was shown in Table 5, a frequency of (il)legal psychoactive substances (ab)use could not be used as a predictor of NPS awareness among Serbian future HCPs. However, in the binarized negative binomial model (the student answers yes if they had tried/used the substance and no if they had never tried it), students who had tried/used marijuana knew 52.6% more NPS names than those who had never had experience with cannabis (IRR: 1.526, CI: 0.953–2.445, *p* = 0.049) (Table 5).

Since only two respondents had ever tried NPSs, the binary logistic regression model was not statistically significant (*p* = 0.226 for socio-demographic and *p* = 0.222 for prior habits as predictors) and does not improve the prediction of NPS use among future HCPs.

Of the 490 students surveyed, only two (0.41%) reported having ever used any NPSs in their lifetime. The main reasons for such behavior were improving concentration and learning ability. Twenty-five students (5.10%) reported that they had been offered NPSs, mostly in the clubs (13.52%), by friends (11.44%) or via the internet (one student, 0.20%). Taking into account the influence of socio-demographic factors, a statistically significant difference was observed only regarding respondents’ sex—more males reported being offered NPS (12% males vs. 3.2% females, *p* = 0.01). Twenty-eight students (5.71%) reported that they had been acquainted with NPS users, and more males reported such acquaintance (11.2% males vs. 4.4% females, *p* = 0.01). In addition, acquaintance with NPS users were more common for respondents who tried/consumed marijuana (*p* = 0.016).

Half of the sample (49.59%) thought that “*The usage of NPS is legal in Serbia*”, while the most common attitude regarding five out of six claims was “*do not know*” (Table 6).

Future HCPs in Serbia advised improvement of education in schools and faculties about NPSs (34%), better legislation (27%), media (23%), and parents’ education (16%) as main targets for protective measures against NPS use in vulnerable populations.

## 4. Discussion

This study examined awareness and attitudes related to NPS as well as the prevalence and predictors of this behavior among future HCPs in Serbia. The socio-demographic and academic characteristics of students as well as their habits regarding (il)licit psychoactive substances (ab)use were evaluated as the main correlates of NPS awareness and use in this study group consisting of 490 future Serbian HCPs.

Even though half of the respondents (54.29%) were tobacco consumers and almost all (95.31%) drank alcohol beverages, such habits could not be used as a predictor of NPS awareness among Serbian future HCPs (IRR = 1.038, CI = 0.659–1.633, *p* = 0.873 and IRR = 1.087, CI = 0.515–2.295, *p* = 0.828, respectively). On the contrary, in the most recent study investigating knowledge and perceptions of synthetic cannabinoids among university students in Jordan, the authors reported that being a smoker (*aOR* = 1.369, 95% Cl = 11.041–1.871, *p* = 0.026) as well as an alcohol user (*aOR* = 2.134, 95% CI = 1.362–3.346, *p* = 0.001) were predictors of good knowledge [13].

The lifetime use of cannabis was reported by 24.49% of the respondents in our study, which is in agreement with the earlier study conducted in Belgrade, Serbia, among medical students (34.9%) [21], as well as with the study including prospective doctors at the University of Novi Sad (33.2%) [17]. According to Shafiq et al., 78% of the medical students had no intention of ever using a drug and a predominant anti-drug opinion was noted among them [22]. In the study by Khalid et al., 82.7% participants did not justify hashish use under any circumstances and 88.2% of them were medical students [23]. It would be extremely important to obtain data regarding the motivation for starting the consumption of psychoactive substances. The self-medication often lies behind substance abuse. This opens new professional questions on the topic of students’ mental health and the need for early recognition and interventions in this vulnerable category.

Paut-Kusturica et al. also determined that students’ knowledge regarding medical cannabis was in strong correlation with previous marijuana consumption, as previous marijuana users were more knowledgeable about therapeutic and side effects, while students who never consumed marijuana were more aware of possible abuse [17]. Similarly, previous habits regarding marijuana consumption were predictors of NPS awareness in our study—students who had tried/used marijuana knew 52.6% more NPS names than those who had never had experience with cannabis (IRR: 1.526, CI: 0.953–2.445, *p* = 0.049).

Although the majority of students were female, the frequency of previous cannabis use was significantly higher among male students. This finding is compatible with those obtained in the recent study reflecting the medical students’ knowledge regarding medical cannabis use in Serbia [17] but not with the results by Moeller et al., where no gender differences was noticed regarding previous marijuana consumption among pharmacy students [24]. One of the reasons for male predomination might be, as well, a greater propensity of the males for psychoactive substances (including NPS), which has been noted in numerous publications [25,26,27,28,29]. Being male is also responsible for better NPS awareness in our study—female students knew 36.5% less NPS names than their male colleagues (IRR: 0.635, CI: 0.399–1.013, *p* = 0.049). Moreover, this gender distribution can be explained by the lower willingness of men to seek help in case of mental problems and the potential search for self-medication through substance abuse, unlike women, who are more willing to seek help by talking to professionals.

In the present survey, number of NPS names students know was rising 15.9% with each study year—the higher the age, the larger the number of NPS they were aware of (IRR: 1.159, CI: 1.025–1.310, *p* = 0.018). This is in accordance with results by other authors worldwide. Assaf et al. reported that second-year students had significantly lower knowledge regarding psychoactive substances than third- (*p* = 0.004) and fourth-year students (*p* < 0.001), with the differences being in the knowledge of the classification of some drugs, dependence criteria, and alcohol withdrawal [14]. Additionally, Cape et al. reported a significant increase in knowledge scores from 23.4% in the second year to 71.8% in the final year of medical education [30]. In the study by Haddad et al., a comparison was made between grade 10 and 11 high school students, where older students reported more awareness towards substance abuse [31]. Moreover, a total of 33% rated their knowledge of substance use as “very good” and most of these were students in the higher grade. Similarly, in the study by Shafiq et al., seniors (fourth- and fifth-year students) reported fewer benefits of substance use compared to juniors (first, second, and third-year students) [22].

NPS awareness in our study was better in pharmacy students (IRR: 1.926, CI: 1.173–3.163, *p* = 0.010) than in medicine students—in other words, pharmacy students knew 92.6% more NPS names than their peers studying medicine. This finding might imply that students who had additional (extra) pharmacology and toxicology classes, like pharmacy students in Serbia, gained better knowledge of NPS. Even though medical students attend psychiatry lessons as well as emergency medicine clinical practice during their fifth and sixth years of studies, they were not singled out regarding their knowledge about the topic. Moreover, only 1.62% respondents from our study mentioned practical (clinical) lectures as sources of information regarding NPSs, whereas theoretical lectures were underlined by 23.79% of the participants. One third of the sample (34%) advised improvements in education in schools and faculties about NPSs as main protective measures against NPS use in vulnerable populations. It also should be emphasized that participants of our study were mostly uncertain regarding all the offered claims about NPS.

In the study evaluating awareness of last grade medical students of the danger of synthetic cannabinoids [15], the internet (including social media) (48.6%) and pharmacology lectures (40.5%) were identified as the most stated sources of information about NPS. In the same study, students with social media accounts had significantly higher NPS awareness scores (*p* < 0.05) [15]. On the other hand, our results have shown that social media account existence did not affect students’ knowledge about NPSs. Additionally, our findings that parents’ professional qualifications as well as a source of income were not predictors of the students’ awareness of NPSs are not in line with a broader literature indicating that some forms of substance use as well as patterns of such use are more common among students whose parents have university education. The authors agreed that many NPSs are currently “legal” and available for purchase in convenient stores and on the internet, making them easily accessible for young adults, especially those with economic means to purchase [32,33,34]. Even though most of the participants in our study lived in urban areas, residence did not affect students’ awareness of NPS, which is not comparable with the previously published data by Martinotti et al. (2015)—more knowledge of NPS was registered among adolescents in urban areas [20].

The present findings indicate that although almost one third of the sample had heard of NPSs and 5.10% reported being offered NPSs, use among students was very low (0.41%). In the study assessing diffusion, knowledge, and risk awareness of NPS among UK students, 81.8% of the student sample had heard of a substance “referred to as a ‘legal high’“ and 31.4% confirmed having used “legal highs” at least once in their life [35]. However, the sensitive nature of asking respondents about previous use of NPSs could be a reason for misreporting on this topic in our study. While students from the UK emphasized first of all the “enjoyable effects” (55.7%) and the “easy availability” (45.70%) of these substances [35], future HCPs in our study reported taking NPSs to improve concentration and learning ability. Only for 15.7% of respondents from the UK could it be speculated that they had taken NPS due to their different/non-recreational nature such as cognitive enhancers, sleeping pills, and drugs to reduce body weight and/or to enhance sexual performances [35]. The potential of substances to induce psychological disturbance should be considered as well. This could lead to serious consequences for the mental health of young people.

The most commonly marked NPS names were Vanilla Sky (methylenedioxypyrovalerone), mephedrone, and Meow-Meow (mephedrone), recognized by 34.8%, 24.3%, and 19.1% of the participants, respectively. These products, together with synthetic cannabinoids, are available over one decade at the Serbian NPS market and their appearance can be traced back to 2010–2011 [5], so that could be a reason for better students’ awareness.

## 5. Conclusions

Analyzing this topic, we should mention the importance of increasing the awareness of students’ mental health, especially medical students as future health care professionals. Designing preventive programs among this population is very significant. All in all, even though a high number of future HCPs claimed to know what NPSs were, numerous misconceptions were noticed. Students’ awareness regarding NPSs was correlated with previous marijuana consumption, subject of study, as well as age and gender. Moreover, a lack of formal education regarding the topic speaks in favor of the necessity for improving theoretical and clinical lectures about NPSs for prospective HCPs in the context of their future role in the prevention and treatment of NPS overdose and addiction.

## Figures and Tables

**Table 1 ijerph-19-14877-t001:** Socio-demographic and academic characteristics of future HCPs in Serbia.

Characteristics	Number (%)
**Age** (years)	21.57 ± 2.04
**Gender**	
Male	112 (22.86)
Female	376 (76.73)
Unknown	2 (0.41)
**Subject of study**	
Medicine	197 (40.20)
Pharmacy	157 (32.04)
Dentistry	136 (27.76)
**Year of study**	
First	95 (19.39)
Second	80 (16.33)
Third	93 (18.89)
Fourth	94 (19.18)
Fifth	93 (18.89)
Sixth	35 (7.14)
**Repeating a study year (redoublement)**	
Yes	93 (18.98)
No	397 (81.02)
**Parents’ highest educational level**	
Elementary school	10 (2.04)
Secondary school	243 (49.59)
College	62 (12.65)
University	173 (35.31)
**Residence (during studies)**	
Urban (more than 100,000 inhabitants)	395 (80.61)
Average (between 30,000 and 100,000 inhabitants)	44 (8.98)
Rural (less than 30,000 inhabitants)	51 (10.41)
**Place of living**	
With parents	157 (32.04)
In university dormitories	57 (11.63)
In rented apartment	148 (30.20)
In own apartment	125 (25.51)
Unknown	2 (0.41)
**Source of incomes during studies**	
Parents	444 (90.61)
Scholarship	24 (4.90)
Parents + scholarship	18 (3.67)
Employment	1 (0.20)
Unknown	3 (0.61)
**Having a social network profile**	
Yes	476 (97.14)
No	11 (2.24)
Unknown	3 (0.61)
**Grand Total**	490 (100)

**Table 2 ijerph-19-14877-t002:** Future HCPs’ habits regarding (il)legal psychoactive substances (ab)use.

Question	N (%)
**Have you ever consumed tobacco?**	
Yes	266 (54.29)
No	224 (45.71)
**If the answer is yes—How often?**	
Every day (active)	56 (11.43)
Sometimes (occasionally)	94 (19.18)
Stopped (quit)	26 (5.31)
Tried (one or two times during life)	90 (18.37)
**Have you ever drunk energy drinks?**	
Yes	431 (87.96)
No	51 (10.41)
Unknown	8 (1.63)
**If the answer is yes—How often?**	
Regularly (at least once a week)	22 (4.49)
Sometimes (e.g., during exam period)	138 (28.16)
Rarely (a few times during life)	271 (55.31)
**Have you ever used alcohol?**	
Yes	467 (95.31)
No	23 (4.69)
**If the answer is yes—How often?**	
Everyday	8 (1.63)
Sometimes (special occasions)	411 (83.88)
Tried (a few times during life)	47 (9.59)
**Have you ever been drunk?**	
Yes	374 (76.33)
No	116 (23.67)
**If the answer is yes—How often?**	
Regularly (each weekend)	48 (9.80)
Sometimes (a few times during a year)	180 (36.73)
Rarely (a few times during life)	146 (29.80)
**Have you ever used marijuana?**	
Yes	120 (24.49)
No	370 (75.51)
**If the answer is yes—How often?**	
Regularly (twice a month)	9 (1.84)
Sometimes (less than twice a month)	36 (7.35)
Once	56 (11.43)
stopped	19 (3.88)
**Do you have a friend who uses marijuana?**	
Yes	268 (54.69)
No	219 (44.69)
unknown	3 (0.61)
**Have you ever tried some of the below-mentioned substances?**	
Heroin (or other opioids)	0
Cocaine	5 (1.02)
Cocaine + LSD + ecstasy	1 (0.20)
Cocaine + methamphetamine	5 (1.02)
Methamphetamine + ecstasy	12 (2.45)
LSD or other hallucinogens	0
Other	1 (0.20)
Unknown	5 (1.02)
No	461 (94.08)
**Do you know someone who uses above-mentioned substances?**	
Yes	126 (25.71)
No	360 (73.47)
Unknown	4 (0.82)

**Table 3 ijerph-19-14877-t003:** Recognition of NPS names among future HCPs in Serbia.

NPS Name	Responses		Percent of Cases
	N	%	
Mephedrone	28	13.6	24.3
Meow-Meow	22	10.7	19.1
Bubbles	9	4.4	7.8
M-Cat	5	2.4	4.3
Phenethylamines	4	1.9	3.5
Methoxetamine	24	11.7	20.9
Desomorphine	20	9.7	17.4
Methylenedioxypyrovalerone	3	1.5	2.6
Ivory Wave	19	9.2	16.5
Vanilla Sky	40	19.4	34.8
Pure Ivory	1	0.5	0.9
Purple Wave	8	3.9	7.0
Charge+	2	1.0	1.7
Ocean Burst	2	1.0	1.7
Sextacy	11	5.3	9.6
Synthetic cannabinoids	4	1.95	3.5
Spice	4	1.95	3.5
Total	206	100.00	179.1

**Table 4 ijerph-19-14877-t004:** Socio-demographic and academic characteristics as predictors of NPS awareness among Serbian future HCPs—negative binomial model (*p*-value) and incidence rate ratios (IRR) with 95% confidence intervals (CI).

Characteristic	Parameter	IRR (CI)	*p*
Gender	male		
	female	0.635 (0.399–1.013)	0.049 *
Age		1.159 (1.025–1.310)	0.018 *
Subject of study	medicine	reference	reference
	pharmacy	1.926 (1.173–3.163)	0.010 *
	dentistry	1.041 (0.623–1.738)	0.879
Repeating a study year	no	reference	reference
	yes	0.712 (0.375–1.354)	0.301
Parents’ highest educational level	secondary and elementary school	reference	reference
	college	0.755 (0.392–1.455)	0.401
	university	1.169 (0.754–1.811)	0.486
Residence	rural	reference	reference
	urban	1.597 (0.765–3.336)	0.213
	average	1.378 (0.510–3.718)	0.527
Place of living	with parents	reference	reference
	in university dormitories	0.647 (0.308–1.360)	0.250
	in rented apartment	0.913 (0.553–1.508)	0.723
	in own apartment	0.747 (0.435–1.282)	0.290
Incomes	parents	reference	reference
	parents + scholarship	1.252 (0.471–3.328)	0.652
	scholarship	0.793 (0.288–2.184)	0.654
Social network profile	no	reference	reference
	yes	/	0.975

* statistical significance.

**Table 5 ijerph-19-14877-t005:** Habits regarding (il)legal psychoactive substances (ab)use as predictors of NPS awareness among Serbian future HCPs—negative binomial model (*p*-value) and incidence rate ratios (IRR) with 95% confidence intervals (CI).

Characteristic	Parameter (Frequency)	IRR (CI)	*p*	Parameter (Binarized)	IRR (CI)	*p*
Smoking	never	reference	reference	no	reference	reference
	everyday (active)	1.497 (0.734–3.052)	0.267	yes	1.038 (0.659–1.633)	0.873
	sometimes (occasionally)	1.119 (0.622–2.012)	0.708			
	stopped (quit)	0.896 (0.339–2.372)	0.826			
	tried	1.380 (0.793–2.400)	0.254			
Energy drinks	never	reference	reference	no	reference	reference
	regularly	0.782 (0.244–2.511)	0.680	yes	0.855 (0.438–1.667)	0.664
	sometimes	0.966 (0.456–2.048)	0.929			
	rarely	0.813 (0.411–1.611)	0.554			
Alcohol	no	reference	reference	no	reference	reference
	yes	1.321 (0.674–2.588)	0.417	yes	1.087 (0.515–2.295)	0.828
Marijuana	never			no	reference	reference
	regularly	1.751 (0.473–6.478)	0.401	yes	1.526 (0.953–2.445)	0.049 *
	sometimes	1.155 (0.530–2.518)	0.716			
	tried	1.452 (0.791–2.668)	0.229			
	stopped (quit)	2.259 (0.874–5.837)	0.092			

* statistical significance.

**Table 6 ijerph-19-14877-t006:** Attitudes of future HCPs in Serbia towards NPSs.

Claims about NPS	Yes N (%)	Do Not Know N (%)	No N (%)
Their usage is legal in Serbia	54 (11.02)	193 (39.39)	243 (49.59)
Their chemical structure is constantly changed (improved).	192 (39.18)	290 (59.18)	8 (1.63)
Their psychoactive action is unpredictable and they can even cause death.	229 (46.73)	251 (51.22)	10 (2.04)
They are easily detectable in blood and urine.	156 (31.84)	287 (58.57)	46 (9.39)
Their intermittent usage is not dangerous.	28 (5.71)	259 (52.86)	202 (41.22)
The users of NPS are obligatory dependent on other illegal psychoactive substances.	153 (31.22)	293 (59.79)	44 (8.98)

## Data Availability

All data are available by reasonable request.

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
