# Peer review of "New Psychoactive Substances: Awareness and Attitudes of Future Health Care Professionals in Serbia"

_ijerph, 2022, doi:10.3390/ijerph192214877_

Round 1
Reviewer 1 Report
Language and style need to be improved:
Line 32: , and
Line 33: , and
Line 33: represents
Line 35: policies
...
Reviewer 2 Report
Review on the manuscript of Jovin VM et al.: “New psychoactive substances: Awareness and attitudes of future health care professionals in Serbia”.
In this manuscript, the authors report the conclusions on a cross-sectional survey conducted on a sample of students of integrated academic studies of medicine, pharmacy and dentistry at the Faculty of Medicine, University of Novi Sad, Serbia, to evaluate prospective health care professionals’ awareness and attitudes regarding new psychoactive substances (NPS). They concluded that: (1) NPS awareness was better in pharmacy students than in medicine students; (2) pharmacy students recognized 92.6% more NPS names than their peers studying medicine; (3) Female students knew 36.5% less NPS names than their male colleagues; (4) the higher the students’ age, the larger the number of NPS they were aware of; and (5) students who had used marijuana knew 52.6% more NPS names than those who had never had experience with cannabis
The study seems to be well designed to answer the main points. The data shown in the manuscript seem to be clear and well explained. However, some issues arise to me, which are listed below for consideration of the authors.
1 - The authors found very interesting data on NPS knowledge by students of pharmacy, medicine, and dentistry. Although the study was designed for those students, do the authors have any idea about the perception of NPS abuse by students from other distinct areas (mathematics, arts…)?
2 - Do the authors believe that the students, as future HCPs, are aware on the consequences of NPS abuse?
3 - What is the reality of NPS awareness in other countries around Servia? And the scenario in the occidental Europe/USA, it is the same?
Reviewer 3 Report
The manuscript is clear and concise. It might be interesting for the readers of the Journal. All parts of the manuscript are well written, however the discussion part should be improved by more in depth interpretations of the results. Also APA style recommendations concerning the presentation of numbers in the paper are not followed.
